# Malignant Melanoma of the Tongue: A Scoping Review

**DOI:** 10.3390/life15020191

**Published:** 2025-01-28

**Authors:** Antonio Di Guardo, Alvise Sernicola, Carmen Cantisani, Steven Paul Nisticò, Giovanni Pellacani

**Affiliations:** 1Dermatology Unit, Department of Clinical Internal Anesthesiological and Cardiovascular Sciences, “Sapienza” University of Rome, 00161 Rome, Italy; diguardo.antonio96@gmail.com (A.D.G.); carmen.cantisani@uniroma1.it (C.C.); steven.nistico@uniroma1.it (S.P.N.); pellacani.giovanni@uniroma1.it (G.P.); 2Dermatology Unit, Department of Medicine (DIMED), University of Padua, 35121 Padova, Italy

**Keywords:** tongue melanoma, mucosal melanoma, oral neoplasms, malignant melanoma, review

## Abstract

Malignant melanoma of the tongue is a rare and highly aggressive neoplasm, constituting less than 2% of oral melanomas. Due to its rarity and atypical clinical presentation, diagnosis and management pose significant challenges. This study provides a scoping review of research on melanoma of the tongue to determine the available data on the epidemiology, clinical features, histopathological characteristics, treatment strategies, and outcomes of this malignancy. Our literature search identified papers published from 1941 to 2024, and 47 individual cases were analyzed. The mean age at diagnosis was 58.6 years, with a male predominance (58.1%). Lesions were most frequently located on the body and lateral borders of the tongue. A high percentage (38.5%) presented with distant metastases at diagnosis, commonly involving the lungs and brain. Histopathological examination highlighted spindle cell morphology in many cases, with immunohistochemical markers such as HMB-45 and S-100 proving essential for diagnosis. Wide local excision with or without neck dissection was the primary treatment, though recurrence rates remained high (20.5%). Despite aggressive management, overall outcomes were poor, reflecting the melanoma’s advanced stage at diagnosis in most cases. This scoping review underscores the need for heightened clinical suspicion, particularly for pigmented or ulcerative lesions of the tongue. Early diagnosis, multidisciplinary management, and further research into the genetic and molecular mechanisms underlying tongue melanoma are crucial to improve outcomes for this rare and aggressive disease.

## 1. Introduction

Primary mucosal malignant melanoma of the head and neck (HNMM) is a rare and aggressive neoplasm, with the tongue being an especially uncommon site for its occurrence. HNMM arises from melanocytes located in the basal layer of the mucosa across various anatomic sites [1]. Unlike cutaneous melanoma, which predominantly develops in the epidermis and is linked to ultraviolet (UV) radiation exposure, mucosal melanoma differs significantly in its epidemiology, genetic profile, clinical behavior, and response to therapy [1,2]. It is not induced by UV radiation and is associated with a poorer prognosis and lower response rates to immunotherapy compared to cutaneous melanoma. HNMM can involve the whole aerodigestive tract, including the oral cavity, the pharyngo-larynx, the nasal fossa, and the paranasal sinuses. Mucosal malignant melanomas of the head and neck most commonly originate in the sinonasal region, accounting for up to two-thirds of cases [3]. These tumors primarily arise from the lateral nasal wall and nasal septum. Melanomas of the oral cavity, which constitute approximately a quarter of cases, frequently affect the palate or gingiva, are often diagnosed at advanced stages, and are accompanied by lymph node metastases. Histologically, these melanomas exhibit diverse growth patterns, including spindled, perivascular, and solid forms [2,3]. Oral melanomas are often asymptomatic until advanced stages, as patients rarely examine their oral cavity closely. They are usually diagnosed after significant swelling, tooth mobility, or bleeding prompts medical attention. Lesions can range from 1.0 mm to over 1.0 cm, with reports of pre-existing pigmented lesions suggesting undiagnosed radial growth-phase melanomas [4]. Treatments typically involve surgery, interferon alpha-2b, immune checkpoint inhibitors, BRAF and MEK inhibitors, and radiotherapy. Multimodal therapy provides the best chance for relapse-free survival compared to single-treatment approaches [5]. The prognosis is poor, with a 5-year survival rate of 10–25%. Early detection and radical surgical excision may significantly improve outcomes. Oral melanoma lacks most equivalent histological landmarks of cutaneous melanoma; instead, tumor thickness or volume serves as a prognostic indicator. Late-stage diagnosis often correlates with extensive tumors and metastases, with most patients succumbing to the disease within 2 years. Recurrence and metastasis are common post surgery. Survival rates vary widely in the literature, ranging from 4.5% to 48%, with most data clustering between 10 and 25% [6]. Other prognostic factors include pigmentation levels and cell type, with non-epithelioid cell types and strong pigmentation associated with better survival outcomes. While oral malignant melanoma represents less than 1% of all melanomas, the incidence of tongue melanoma is even rarer, constituting less than 2% of oral and nasal melanomas [7,8]. This rarity poses significant challenges in terms of diagnosis, treatment, and prognosis. The asymptomatic nature of early-stage tongue melanomas often leads to late detection, by which time the malignancy has advanced, making radical treatment difficult to achieve [9]. Primary melanoma of the tongue typically arises from melanocytes present in the oral mucosa, but the factors contributing to its development, as well as the etiological pathways involved, are not clearly established. Some studies suggest the potential role of environmental factors, such as tobacco use, alcohol consumption, and chronic irritation, although these associations remain speculative [10]. Recent genetic investigations have identified alterations in C-KIT signaling and mutations in genes such as BAP1, highlighting a potential genetic predisposition in certain cases, but further research is needed to confirm these findings [11,12] (Figure 1).

Clinically, melanoma of the tongue often presents as a pigmented lesion, although non-pigmented (amelanotic) forms can occur, complicating early detection [13,14]. Amelanotic mucosal melanoma displays diverse histological patterns, including epithelioid, spindle, pleomorphic, small cell, and plasmacytoid variants. The absence of melanin pigment complicates diagnosis [15]. Differential diagnoses include small blue cell tumors (e.g., Ewing’s sarcoma/primitive neuro-ectodermal tumor (PNET)), olfactory neuroblastoma, neuroendocrine carcinoma, sinonasal undifferentiated carcinoma, poorly differentiated squamous cell carcinoma, large B-cell lymphoma, and sarcoma-like spindle cell tumors [16]. The palate and gingiva are common sites of oral melanoma, but when melanoma occurs on the tongue, it typically involves the lateral borders or ventral surface. Due to the complex anatomy and rich vascularization of the tongue, this type of melanoma exhibits a particularly aggressive behavior, with a high likelihood of metastasis to regional lymph nodes and distant organs, contributing to a poor prognosis [17]. The treatment for tongue melanoma is primarily surgical, often involving wide excision of the tumor along with neck dissection to manage lymphatic spread [17,18]. However, despite aggressive surgical intervention, recurrence rates are high, and the overall survival outcomes remain poor.

A closely related entity is clear cell sarcoma (CCS), a rare and aggressive tumor primarily affecting young adults, often arising in the deep soft tissues of the lower extremities [19]. CCS of the tongue differs from melanoma in its presentation, typically appearing as a rapidly growing, mobile nodular mass or a submucosal swelling [20]. While these two entities share similarities, such as expression of melanoma markers, CCS lacks BRAF mutations and is characterized by distinct genetic features, including EWSR1–ATF1 or EWSR1–CREB1 gene fusions. A rarer variant, CCS of the gastrointestinal (GI) tract, exhibits unique histological and immunohistochemical profiles, leading some researchers to propose it as a distinct entity [21]. Cases of CCS in the head and neck region are exceedingly rare, with only a handful documented in the literature.

Given the rarity of melanoma of the tongue and the paucity of comprehensive data on its epidemiology, clinical presentation, and treatment outcomes, a scoping review is warranted to collect the available literature on tongue melanoma, focusing on its prevalence, risk factors, clinical features, and treatment strategies. By synthesizing these findings, our study aims to identify existing gaps in the current body of knowledge to aid the planning of future research to improve the diagnosis and management of this rare and challenging condition.

## 2. Materials and Methods

This study provides a scoping review of research on malignant melanoma and clear cell sarcoma of the tongue to determine what the available data on the epidemiology, clinical features, histopathological characteristics, treatment strategies, and outcomes of this malignancy are. Studies were selected based on predefined inclusion and exclusion criteria. We included case reports, case series, and review articles that provided information on the clinical presentation, histopathology, treatment, or outcomes of tongue melanoma. To ensure inclusivity, case reports with incomplete data were also included, recognizing the rarity of this condition and the need to synthesize as much evidence as possible. Studies were excluded if they lacked relevance to tongue melanoma, or did not provide clinical or histopathological data. This study was conducted following the Preferred Reporting Items for Systematic Reviews and Meta-Analyses (PRISMA) extension for scoping reviews (PRISMA-ScR) [22]. This review adheres to the framework of a scoping review, focusing on mapping the existing evidence on tongue melanoma. Consistent with the objectives of scoping reviews, no formal critical appraisal of sources was carried out to provide a broader overview of the existing evidence [23,24]. The included studies consisted primarily of case reports and small case series, which inherently carry methodological limitations, including selection, reporting, and publication biases. To ensure transparency and methodological rigor, the PRISMA-ScR checklist has been provided as a Appendix A [22]. Reference sections were meticulously examined to ensure the inclusion of all relevant reports.

A literature search was conducted from 1941 (year of the first documented case) [25] to 2024 (literature search performed on 11 November 2024), encompassing the PubMed, EMBASE, and Cochrane CENTRAL databases. The search terms “melanoma of the tongue”, “tongue melanoma”, and “melanoma AND tongue” were employed across all databases. A detailed search strategy for PubMed used the following first-step search criteria: (((“Melanoma”[Mesh] OR melanoma*[Title/Abstract]) AND (“Tongue”[Mesh] OR tongue[Title/Abstract] OR lingual[Title/Abstract])) OR ((“Clear Cell Sarcoma”[Mesh] OR “clear cell sarcoma”[Title/Abstract] OR “melanoma of soft parts”[Title/Abstract]) AND (“Tongue”[Mesh] OR tongue[Title/Abstract] OR lingual[Title/Abstract]))) AND (“Humans”[Mesh] AND (“Case Reports”[Publication Type] OR “Review”[Publication Type] OR “Epidemiology”[Mesh] OR “Diagnosis”[Mesh] OR “Therapeutics”[Mesh] OR “Prognosis”[Mesh])). The second-step search criteria were used as follows: ((“Tongue Neoplasms”[Mesh] OR “Tongue”[Mesh]) AND (“Melanoma”[Mesh] OR “Clear Cell Sarcoma”[Mesh]) AND (malignant[Title/Abstract] OR cancer[Title/Abstract] OR neoplasm[Title/Abstract])).

The process of selection of the sources of evidence was performed by two authors (A.D.G. and A.S.) that independently assessed the titles, abstracts, and text of studies identified by the search; results were compared and discrepancies were resolved by discussions, including a third author (C.C.) when needed. Then, the charting of data from the included sources was performed independently by two reviewers (A.D.G. and A.S.) using a standardized data-charting template to capture variables such as patient demographics, clinical presentation, histopathology, treatment modalities, and outcomes. While efforts were made to include comprehensive data, many studies lacked long-term follow-up information, precluding insights into overall survival and the use of second-line or alternative therapeutic approaches. In several cases reported by pathologists, only clinical descriptions and diagnoses were provided, with no data on follow-ups or treatment outcomes. All statistical analyses were conducted using R version 4.0.2 and RStudio version 1.2.5033 (The R Foundation for Statistical Computing, Vienna, Austria), with descriptive statistics, including frequencies and percentages, computed for all variables.

## 3. Results

This review analyzed 47 cases of tongue melanoma and CCS derived from 47 distinct publications. The selection process began with the identification of 360 records through database searches and an additional 3 records identified via references. After removing duplicates, 342 records underwent screening. Of these, 263 were excluded at the screening stage. For the eligibility phase, 79 full-text articles were assessed. A total of 32 were excluded for the following reasons: 8 were reviews or meta-analyses, 15 were unrelated to the topic, and 9 were unavailable. Ultimately, 47 studies were included, each contributing one unique case (Figure 2).

The study collected data on paper characteristics (first author, year, and number of cases reported) and on demographic and clinical characteristics (age, sex, site, clinical morphology, time to treatment of the primary lesion, recurrence or metastasis, therapy, and histology). A comprehensive summary of the demographic, clinical, and histological characteristics of patients with tongue melanoma is presented in Appendix A, while Appendix A reports the main epidemiological, clinical, and histological features of the included cases from the literature [13,14,17,18,19,20,25,26,27,28,29,30,31,32,33,34,35,36,37,38,39,40,41,42,43,44,45,46,47,48,49,50,51,52,53,54,55,56,57,58,59,60,61,62,63,64].

### 3.1. Demographics

Out of the 43 cases for which sex was specified, 25 (58.14%) were male and 18 (41.86%) were female, resulting in a male-to-female ratio of 1.38:1. Four cases did not specify sex. The mean age at diagnosis was 58.6 years, with a median age of 62 years. The age range spanned from 7 to 90 years. Among the 40 cases with specified age ranges, the most common age group at diagnosis was 65 to 78 years, which included 14 cases (35%) out of the total, followed closely by the 49 to 64 years group, with 13 cases (32.5%). Younger patients, specifically those between the ages of 19 to 33 years, represented a small fraction, with only five cases (12.5%) in this range. The incidence in children and young adults was exceedingly low, with no cases reported in the 0 to 5 or 14 to 18-year categories, and only one case (2.5%) in the 6 to 13-year group. The time interval between clinical diagnosis and treatment of the primary lesion ranged between 1 and 108 months, with a median of 5 months and an average of 18.3 months (Table 1).

### 3.2. Anatomical Location and Size

The anatomical location of the tongue melanoma varied, with the majority of lesions affecting the body of the tongue (Figure 3). Specifically, 35 of the 37 cases (94.59%) had lesions located on the body of the tongue. These included lesions on the dorsum, lateral borders, and base. A total of 12 cases (32.43%) were located on the dorsum, an equal number of cases (32.43%) were located on the lateral borders, and 11 cases (29.72%) were found on the base of the tongue. Two cases (5.4%) affected the apex of the tongue. In a small number of cases, multiple sites were involved, with three cases reporting lesions in more than one location. The data regarding the anatomical location were unavailable for 10 cases.

The maximum diameter of the melanoma lesions varied widely, with the median diameter measuring 2.69 cm, with a range from 0.4 to 6.2 cm. Among the 31 cases reporting lesion size, the distribution was as follows: 1 case (3.22%) had a lesion ≤ 0.5 cm, 2 cases (6.45%) had lesions between 0.6 and 0.9 cm, and 7 cases (22.58%) had lesions ranging from 1.0 to 1.9 cm. The most common size range was between 3.0 and 3.9 cm, with eight cases (25.8%) falling within this category. A similar proportion of cases, eight (25.8%), had lesions ≥ 4.0 cm in size. In total, 51.6% of the lesions were larger than 3 cm, suggesting that a significant proportion of tongue melanomas are relatively large at diagnosis. Possible associated habits or trauma were sparsely recorded in the published cases from the literature; therefore, correlations with the location and size of lesions could not be hypothesized (Table 2). Notably, out of the documented cases, three out of four patients who smoked and two out of three patients who consumed alcohol experienced either mortality or metastatic disease progression.

### 3.3. Clinical Morphology

The clinical presentation of tongue melanoma was varied, though papules and nodules were the most common forms, observed in 27 of the 33 cases (81.81%) with clinical morphology information available. Macules or patches were less frequent, seen in three cases (9.09%), while submucosal masses and plaques were observed in two cases (6.06%) and one case (3.03%), respectively. Reported secondary morphological features included black pigmentation, noted in 21 cases (63.64%), followed by ulceration and bleeding, seen in 11 cases (33.33%). Amelanotic melanomas, lacking the typical dark pigmentation, were seen in six cases (18.18%). Additionally, satellite lesions were reported in two cases (6.06%), and one case (3.03%) showed a cystic appearance. One case (3.03%) was associated with leukoplakia. “Giant” lesions exceeding 6 cm in size were reported in two cases (5.43%).

### 3.4. Histopathology

Though all cases were histologically confirmed (melanoma in 43 cases and CCS in 4), a description of histopathological findings was reported for 23 cases. Spindle cell morphology, a common feature of melanoma, was found in eight cases (34.78%), while three cases (13.04%) exhibited epithelioid cell morphology. The remaining cases showed a combination of spindle and epithelioid cell morphology, seen in eight cases (34.78%). The infiltration of underlying tissues was noted in 11 of the 23 cases (47.83%), while 3 cases (13.04%) demonstrated a pagetoid spread, a phenomenon where cancer cells spread along the surface layer of the epithelium. Vascular or perineural invasion was reported in four cases (17.39%). Immunohistochemistry findings were available for 17 cases. The most frequently positive markers included HMB45 (13/17), S100 (11/17), SOX-10 (4/17), MART-1/Melan-A (3/17), and vimentin (3/17).

### 3.5. Metastasis and Recurrence

Metastasis at the time of diagnosis was reported in 15 of the 39 cases (38.46%), with distant metastasis to the lung and pleura being the most common (6 cases). Other metastatic sites included the brain (four cases), the liver (one case), the spinal cord (one case), and the thyroid (one case). Nodal metastasis was observed in 14 cases (35.9%). In 10 cases (25.64%), there was no metastasis at the time of diagnosis, and metastasis staging was unknown in 8 cases. Local recurrence of melanoma occurred in eight cases (20.51%), and local invasion was observed in five cases (12.82%). One of the cases showed CCS features, and one was diagnosed as epithelioid melanoma. A significant number of tumors showed spindle cell features, observed in eight cases (34.78%). Furthermore, local recurrence was treated with either surgery and radiation therapy (RT) or RT alone, depending on the clinical circumstances.

### 3.6. Treatment and Outcomes

Treatment data were available for 39 cases, and most patients underwent wide local excision (WLE) as part of their initial treatment (Table 3). In 13 cases (33.33%), WLE was combined with neck dissection. In seven cases (17.95%), WLE was performed with additional chemotherapy (CT) or RT. Other treatment modalities included interferon alfa-2b (three cases), anti-PD-1/PD-L1 therapy (three cases), and imatinib (one case). Palliative therapies, such as interferon alfa-2b and laser treatments, were used in some cases.

The clinical outcome for these patients varied (Table 4). Of the 31 cases with outcome data, 9 (29%) had a clear outcome with no progression. Twelve patients (38.46%) died due to tongue melanoma, while two (6.45%) died from other causes. Four patients (12.9%) showed worsening in their condition, and one patient (3.23%) was lost to follow-up. Local recurrence occurred in several cases, with treatment including surgery, RT, and CT.

## 4. Discussion

Tongue melanoma is a rare and aggressive malignancy constituting less than 2% of oral melanomas, presenting unique challenges in diagnosis, management, and prognosis. Conley, in a 1974 review of 52 cases, commented on the absence of melanoma occurrences in the tongue, underscoring the rarity of this presentation [65]. However, as highlighted by Batsakis, at least 25% of mucosal melanomas can mimic benign lesions, making clinical diagnosis particularly challenging [66]. This observation emphasizes the critical need to consider mucosal melanoma in the differential diagnosis of head and neck neoplasms and to maintain a high index of suspicion, even for lesions that appear clinically insignificant. The base of the tongue is a frequent site of metastases from cutaneous melanomas originating in other locations. Such metastatic involvement often signals poor prognosis and may manifest many years after the initial melanoma diagnosis [64]. Furthermore, amelanotic lesions of the tongue can resemble squamous cell carcinoma, complicating accurate and timely diagnosis [67]. These factors highlight the necessity for vigilance in the evaluation of head and neck lesions, especially in patients with a history of melanoma or atypical presentations.

This scoping review synthesizes data from 47 cases to provide a comprehensive summary of the demographic, clinical, histopathological, and treatment-related aspects of this tumor. Epidemiologically, tongue melanoma demonstrates a slight male predominance, with a mean age of 58.6 years at diagnosis. This aligns with the observation that mucosal melanomas primarily affect older adults, possibly due to cumulative genetic or environmental exposures. Anatomically, lesions frequently arise on the lateral borders, dorsum, and base of the tongue, sites characterized by rich vascularization, which may facilitate early metastasis [67].

### 4.1. Pathophysiology of Tongue Melanoma

The etiology of tongue melanoma remains poorly understood, but environmental and genetic factors may contribute to its development. Unlike cutaneous melanoma, where UV radiation is a well-established risk factor, tongue melanoma lacks a definitive causative link to external environmental exposures. Nevertheless, tobacco use, alcohol consumption, and chronic irritation from prostheses or poor oral hygiene have been suggested as potential contributors [17,68]. While these associations are plausible, the current evidence is insufficient to establish causality. In this review, patients with tongue melanoma associated with habits such as smoking and alcohol consumption may have worse outcomes, as suggested by the high rates of mortality or metastatic progression in these cases. However, the limited data available and the small sample size make it impossible to confirm this hypothesis at present. The role of trauma, including previous surgical or radiation treatment, in the development of mucosal melanomas is also debated, as isolated reports describe melanoma arising at sites of chronic irritation, but larger studies are lacking [40,58,69]. Genetic predisposition may play a significant role in the pathogenesis of tongue melanoma. Recent studies on oral malignant melanomas have identified mutations in tumor suppressor genes such as BAP1, as well as alterations in pathways involving p16 and p53 [45]. A loss of pRb2/p130 expression has been associated with more aggressive diseases and poorer outcomes, whereas its presence may correlate with earlier malignancy stages [45]. Although these findings highlight the importance of genetic alterations, the molecular mechanisms driving tongue melanoma remain poorly understood compared to cutaneous forms. Further studies are needed to explore these genetic pathways and identify therapeutic targets, especially for advanced or refractory cases.

### 4.2. Clinical Presentation and Diagnosis

Clinically, the presentation of tongue melanoma is heterogeneous, ranging from pigmented nodules or plaques to amelanotic lesions that mimic other malignancies, such as squamous cell carcinoma. This variability underscores the importance of histopathological confirmation, particularly given that up to 18% of cases are amelanotic and lack the pigmentation typically associated with melanoma. Immunohistochemical markers such as HMB-45, S-100, and SOX-10 play a critical role in distinguishing melanoma from other oral tumors, with spindle and epithelioid cell morphologies frequently observed in histopathological examinations. The aggressive behavior of tongue melanoma is reflected in its high rates of metastasis and recurrence. Metastatic involvement at diagnosis was reported in 38.5% of cases, most commonly affecting the lungs and brain, while regional lymph node metastases occurred in 35.9% of patients. These findings suggest that tongue melanoma often presents at an advanced stage, contributing to its poor prognosis. The five-year survival rate for mucosal melanomas, including those of the tongue, remains dismally low at 6.6–20%, which is significantly worse than that of cutaneous melanoma [60]. The advanced stage at presentation is attributed to the tumor’s asymptomatic nature in its early stages, its hidden location, and the difficulty in achieving early diagnosis [70]. Several factors have been linked to poor prognosis, including tumor size, depth of invasion, presence of metastases, histopathological features such as vascular or perineural invasion and the absence of symptoms in the early stages of the disease, and challenges in achieving wide radical excision due to anatomical constraints [60]. Also, large lesions (>3 cm) and those with evidence of satellite nodules or deep tissue infiltration are associated with worse outcomes. In contrast, early-stage lesions confined to the mucosa without metastases have a better prognosis, albeit such cases being rare.

### 4.3. Treatment and Management: Multidisciplinary Care

Treatment primarily involves WLE with or without neck dissection, which remains the cornerstone for local control. However, achieving adequate surgical margins in the anatomically complex and functionally critical region of the tongue is challenging [52]. This often necessitates multimodal approaches, combining surgery with RT, CT, or immunotherapy. Despite these efforts, recurrence rates remain high, with 20.5% of cases experiencing local recurrence post treatment. RT and CT, while frequently used as adjuvant or palliative options, have limited efficacy due to the intrinsic radioresistance of mucosal melanomas [52].

Immune checkpoint inhibitors, particularly anti-PD1/PD-L1 therapies, have transformed the management of advanced melanomas, including mucosal subtypes, by demonstrating superior survival outcomes compared to traditional therapies [58,71]. However, their role in tongue melanoma specifically remains unclear, due to the rarity of this condition and the limited representation of these agents in the reviewed cases. This underrepresentation is partly attributed to the retrospective nature of most studies and their publication before the widespread clinical adoption of these agents. Despite these limitations, the distinct molecular and clinical features of tongue melanoma—such as its high rates of locoregional and distant metastasis—suggest that anti-PD1 therapies could play a pivotal role in improving outcomes. Nevertheless, robust conclusions are hampered by the lack of prospective, tongue-specific studies and the incomplete reporting on subsequent lines of therapy in many cases. Targeted therapies, including BRAF inhibitors (e.g., vemurafenib and dabrafenib) and MEK inhibitors (e.g., trametinib), have shown significant efficacy in BRAF-mutated cutaneous melanomas [72]. However, BRAF mutations are rare in mucosal melanomas, including tongue melanoma, limiting the applicability of these treatments. In contrast, c-KIT mutations are more frequently observed in mucosal melanomas, particularly in the head and neck region [73,74]. KIT inhibitors, such as imatinib, have demonstrated modest efficacy in KIT-mutated cases, with isolated reports of durable responses. The combination of KIT inhibitors with immune checkpoint inhibitors is a potential therapeutic avenue that warrants further exploration, given the promising synergistic effects observed in other melanoma subtypes. Furthermore, emerging strategies such as neoadjuvant and combination immunotherapies (e.g., anti-PD1 with anti-CTLA4) hold promise in managing locally advanced or metastatic mucosal melanomas [49]. While combination therapies have demonstrated improved progression-free survival and response rates in cutaneous melanoma, their benefits in mucosal melanomas, including tongue melanoma, appear less pronounced, and are associated with higher toxicity. The lack of statistically significant differences in comparative analyses between monotherapy and combination regimens for mucosal melanomas, as well as the increased risk of severe adverse effects, suggest that anti-PD1 monotherapy may currently be the most prudent first-line treatment for tongue melanoma [71]. Given the paucity of data specific to tongue melanoma, an urgent need for further research to elucidate the impact of these therapies remains. Prospective studies investigating the molecular profile of tongue melanoma, particularly the predictive value of PD-L1 expression and KIT mutations, are essential to guide the selection of targeted drugs and immunotherapies.

Quality of life considerations are paramount in managing tongue melanoma, due to the functional impairments associated with surgical interventions. Patients often experience difficulties in speech, swallowing, and nutrition, necessitating multidisciplinary approaches involving oncologists, surgeons, speech therapists, and dietitians [49]. Advances in reconstructive techniques, including the use of flaps and prosthetics, have partially mitigated these challenges, although post-operative recovery remains prolonged [59].

### 4.4. Limitations

The scoping methodology of this review carries inherent limitations; charted data were reported in the review irrespective of a critical appraisal of the sources in order to achieve a comprehensive summary of all existing evidence. Consequently, we did not aim to define the quality of the included studies, which consisted of case reports and small case series. Moreover, the paucity of evidence found did not justify undertaking a systematic review. This scoping approach prevented us from performing additional statistics, including an analysis of demographic data or survival. As far as the limitations of the included studies are concerned, reports of isolated cases are prone to selection bias, as they often emphasize rare, severe, or atypical cases, potentially skewing the representation of tongue melanoma’s clinical spectrum. Reporting bias may further affect the findings, as published cases are likely to highlight positive or unusual outcomes, while publication bias may result in the underrepresentation of inconclusive or negative data. Despite these limitations, the review provides a comprehensive synthesis of the existing evidence, offering valuable insights into this rare malignancy. To mitigate these biases in future research, setting up large disease registries or multicenter studies will be essential. Another significant limitation of the included studies was the frequent lack of long-term follow-up data, which restricted our ability to assess overall survival and therapeutic response beyond first-line treatments. This was further complicated by some reports that completely lacked follow-up and treatment details.

## 5. Conclusions

Tongue melanoma, though exceedingly rare, is a malignancy of considerable clinical importance due to its aggressive nature and challenging diagnosis. Its rarity demands heightened awareness among clinicians, as its early-stage presentations often mimic benign or less aggressive lesions, leading to delayed diagnoses. A definitive diagnosis hinges on meticulous histopathological examination supported by immunohistochemical markers, which are essential for distinguishing tongue melanoma from other oral neoplasms. Despite advances in oncology, the prognosis for tongue melanoma remains poor, with high rates of recurrence and metastasis. Traditional therapies such as radiotherapy and chemotherapy show limited efficacy, while novel immunotherapies, particularly PD-1/PD-L1 inhibitors, hold promise but require further specific validation for mucosal melanomas. To address these challenges, a comprehensive and multidisciplinary strategy is essential. This includes not only medical and surgical interventions, but also supportive care, to manage the functional impairments commonly associated with treatment, such as difficulties in speech and swallowing. Advances in reconstructive techniques and postoperative care have the potential to significantly enhance patients’ quality of life. Finally, basic research to unravel the underlying genetic and molecular mechanisms of tongue melanoma will be critical to improve our understanding of the pathophysiology of this rare entity, bridging the current knowledge gap.

## Figures and Tables

**Figure 1 life-15-00191-f001:**
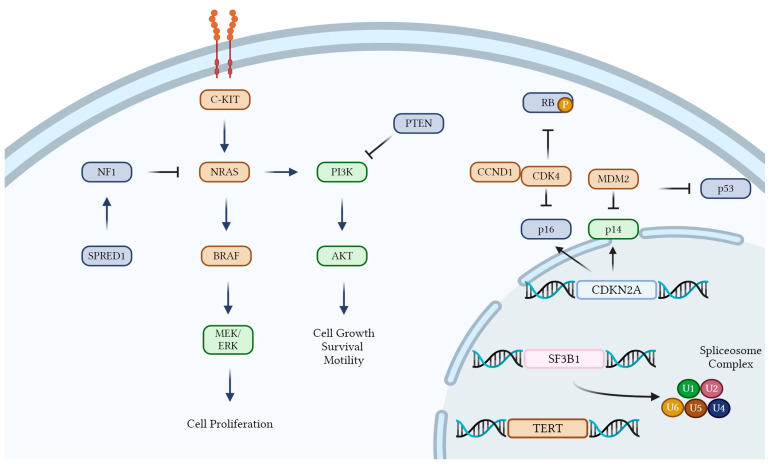
Molecular pathways contributing to the progression of mucosal melanoma. Red-filled rectangles represent genes with activating mutations or amplifications, while blue-filled rectangles denote genes with inactivating mutations or deletions. Created in BioRender. Di Guardo, A. (2025) https://BioRender.com/j31o697 (accessed on 19 January 2025).

**Figure 2 life-15-00191-f002:**
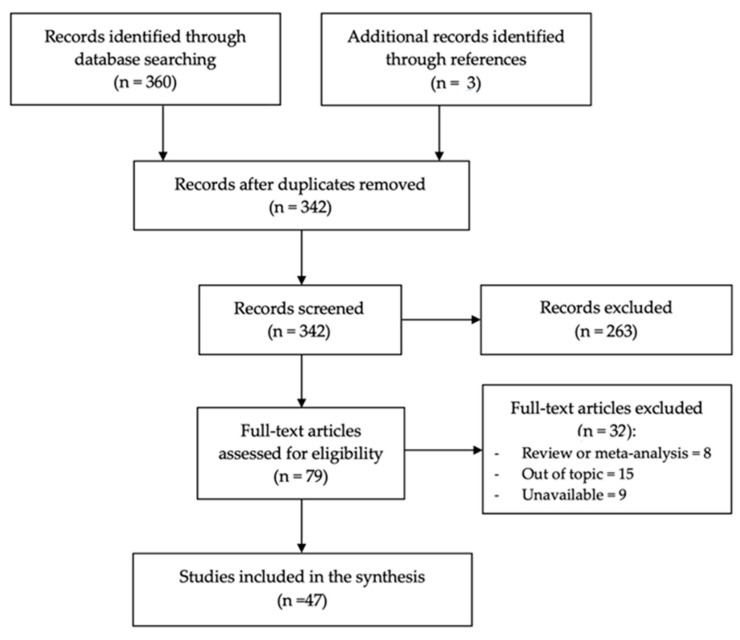
Flow chart of search results and study selection according to the Preferred Reporting Items for Systematic Reviews and Meta-Analyses (PRISMA) extension for scoping reviews (PRISMA-ScR).

**Figure 3 life-15-00191-f003:**
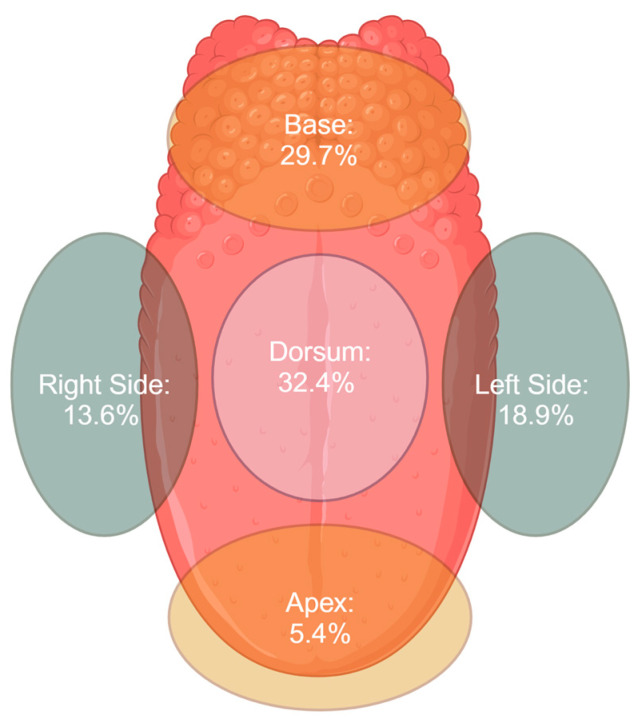
Frequency of different anatomical locations for melanoma of the tongue. Created in BioRender. Di Guardo, A. (2025) https://BioRender.com/q01j455 (accessed on 19 January 2025).

**Table 1 life-15-00191-t001:** Time elapsed from diagnosis to treatment of the primary lesion in tongue melanoma.

Interval (n = 24)	Frequency (%)
≤6 months	15 (62.5%)
7–11 months	1 (4.2%)
≥12 months	8 (33.3%)
NP	23

Abbreviations: n, number of patients; and NP, not published.

**Table 2 life-15-00191-t002:** Habits and trauma potentially associated with tongue melanoma.

Habits or Previous Trauma (n = 11)		Frequency
Smoking and tobacco chewing		4
Alcohol consumption (n = 3)	Minimal	2
	Moderate	0
	High	1
Trauma or surgery		4

Abbreviations: n, number of patients.

**Table 3 life-15-00191-t003:** Treatment approaches for tongue melanoma.

Disease Target	Treatment	Frequency
Primary Tumor (n = 39)	Wide Local Excision (WLE) ± neck dissection	13
	Simple excision	2
	Surgery (unspecified)	5
	WLE or surgery + CT or RT	7
	WLE + interferon alfa-2b	3
	WLE + anti-PD-1/PD-L1 therapy	2
	WLE + imatinib	1
	Palliative therapy	6
	Interferon alfa-2b	2
	Laser	1
	CT	3
Local Recurrence (n = 5)	Surgery + RT	2
	RT	2
	Surgery	1
Metastasis (n = 6)	CT	3
	CT + RT	1
	No treatment	2

Abbreviations: CT, chemotherapy; n, number of patients; RT, radiation therapy; and WLE, wide local excision.

**Table 4 life-15-00191-t004:** Outcomes in cases of tongue melanoma.

Outcome (n = 31)	Frequency
Clear	9
Death due to tongue melanoma	12
Death due other causes	2
Condition stable	4
Condition worsened (including palliative care)	1
Lost to follow-up	3

Abbreviations: n, number of patients.

## Data Availability

No new data were created or analyzed in this study. Data sharing is not applicable to this article.

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
