# Peer review of "Malignant Melanoma of the Tongue: A Scoping Review"

_life, 2025, doi:10.3390/life15020191_

Round 1
Reviewer 1 Report
Comments and Suggestions for Authors
This paper is missing some more important issues, as follows:
1. Please, define the type of review in this article - scoping or systematic. It is a fundamental information (for more details see: Munn Z, Peters MDJ, Stern C, Tufanaru C, McArthur A, Aromataris E. Systematic review or scoping review? Guidance for authors when choosing between a systematic or scoping review approach. BMC Med Res Methodol. 2018 Nov 19;18(1):143. doi: 10.1186/s12874-018-0611-x. ). Please, add such information to the title of the paper!
2. Survival analysis – is it possible to perform such analysis from the data gathered from the 47 reviewed clinical cases (47 patients)? Could it be possible to create a survival curve for the tongue melanoma?
3. PRISMA guidelines require to present limitations for the scoping review process. Please, add information about such limitations to the paper, otherwise, the article is lacking a fundamental issue.
4. A CATREG (categorical regression with optimal scaling using alternating least squares) analysis could be helpful in analyzing the demographic data in relation to the clinical significance of melanoma. Please try to perform additional advanced statistics to elucidate the hidden “real” relationships between demographic data and clinical significance in this type of melanoma. Descriptive statistics are not enough in analyzing variables from clinical cases.
Author Response
Reviewer 1
Dear Reviewer 1,
Thank you for your careful review of our paper and for your constructive comments. Please find our responses provided below point by point.
This paper is missing some more important issues, as follows:
- Please, define the type of review in this article - scoping or systematic. It is a fundamental information (for more details see: Munn Z, Peters MDJ, Stern C, Tufanaru C, McArthur A, Aromataris E. Systematic review or scoping review? Guidance for authors when choosing between a systematic or scoping review approach. BMC Med Res Methodol. 2018 Nov 19;18(1):143. doi: 10.1186/s12874-018-0611-x. ). Please, add such information to the title of the paper!
Thank you for your comment. It is now clearly stated in the title that this is a scoping review. This is also explained in the text. Please see, in the Introduction: “Given the rarity of melanoma of the tongue and the paucity of comprehensive data on its epidemiology, clinical presentation, and treatment outcomes, a scoping review is warranted to collect the available literature on tongue melanoma, focusing on its prev-alence, risk factors, clinical features, and treatment strategies. By synthesizing these findings our study aims to identify existing gaps in the current body of knowledge to aid the planning of future research to improve the diagnosis and management of this rare and challenging condition.” and, in the Materials: “This study provides a scoping review of research on malignant melanoma and clear cell sarcoma of the tongue to determine what is the available data on the epidemiology, clinical features, histopathological characteristics, treatment strategies, and outcomes of this malignancy.”
- Survival analysis – is it possible to perform such analysis from the data gathered from the 47 reviewed clinical cases (47 patients)? Could it be possible to create a survival curve for the tongue melanoma?
Thank you for your suggestion. It was not possible to perform additional analysis on these data and this limitation to our scoping review is clearly presented in paragraph 4.4 Limitations. Please see “… This scoping approach prevented us from performing additional statistics, including analysis of demographic data or survival.”
- PRISMA guidelines require to present limitations for the scoping review process. Please, add information about such limitations to the paper, otherwise, the article is lacking a fundamental issue.
Thank you for your comment. A relevant limitations paragraph was added according to the PRISMA-ScR extension. Please see 4.4 Limitations: “The scoping methodology of this review carries inherent limitations: charted data was reported in the review irrespective of a critical appraisal of the sources in order to achieve a comprehensive summary of all existing evidence. Consequently, we did not aim to define the quality of the included studies, which consisted in case reports and small case series. Moreover, the paucity of evidence found suggested against undertaking a sys-tematic review. This scoping approach prevented us from performing additional statistics, including analysis of demographic data or survival. As far as the limitations of the in-cluded studies are concerned, reports of isolated cases are prone to selection bias, as they often emphasize rare, severe, or atypical cases, potentially skewing the representation of tongue melanoma's clinical spectrum. Reporting bias may further affect the findings, as published cases are likely to highlight positive or unusual outcomes, while publication bias may result in underrepresentation of inconclusive or negative data. Despite these limitations, the review provides a comprehensive synthesis of the existing evidence, offering valuable insights into this rare malignancy. To mitigate these biases in future research, setting up large disease registries or multicenter studies will be essential. Another significant limitation of the included studies was the frequent lack of long-term follow-up data, which restricted our ability to assess overall survival and therapeutic response beyond first-line treatments. This was further complicated by some reports which completely lacked follow-up and treatment details.”
- A CATREG (categorical regression with optimal scaling using alternating least squares) analysis could be helpful in analyzing the demographic data in relation to the clinical significance of melanoma. Please try to perform additional advanced statistics to elucidate the hidden “real” relationships between demographic data and clinical significance in this type of melanoma. Descriptive statistics are not enough in analyzing variables from clinical cases.
Thank you for your suggestion. We considered that additional statistics were not applicable to our study as they are usually not done in scoping reviews. This limitation to our scoping review is clearly presented in paragraph 4.4 Limitations. Please see “… This scoping approach prevented us from performing additional statistics, including analysis of demographic data or survival.”
Reviewer 2 Report
Comments and Suggestions for Authors
The article “Malignant melanoma of the tongue: a review” is interesting. I will make some comments to improve the manuscript with the intention of avoiding confusion.
Although it is not a systematic review (“a review”), the authors present the methodology as if it were. Although they comment “no formal methods to assess the risk bias in study design”, in line 226 it is defined as a systematic review. If it is a systematic review, I propose:
- Use more recent PRISMA criteria, in which there are some modifications of those referred to (Stroup 2000, Moher 2009), such as the flow chart and the different checklists.
- Define the quality of the studies. If the reviewed articles offer quality related to the guideline for publishing cases (Clinical Case Reporting Guideline)
- Selection of articles with inclusion criteria of melanoma vs clear cell sarcoma diagnosis.
- Include some case of tongue melanoma described in previous case series.
- It would be necessary to include the table of the reviewed articles in the manuscript as well as the corresponding bibliographic references so that readers can keep in mind their characteristics.
In summary,
1) the cases must be presented in a transparent manner and diagnosed with melanoma, completing the entire sample [Spindle cell (8), epitheloid cell (3) mixed (8)....];
2) set objectives and conclusions accordingly.
Thank you
Author Response
Reviewer 2
The article “Malignant melanoma of the tongue: a review” is interesting. I will make some comments to improve the manuscript with the intention of avoiding confusion.
Dear Reviewer 2,
Thank you for your efforts on our paper and for your constructive comments. Please find our responses provided below point by point.
Although it is not a systematic review (“a review”), the authors present the methodology as if it were. Although they comment “no formal methods to assess the risk bias in study design”, in line 226 it is defined as a systematic review. If it is a systematic review, I propose:
Thank you, to avoid a potential source of confusion we have clarified in the title that this is a scoping review. This is also explained in the text. Please see, in the Introduction: “Given the rarity of melanoma of the tongue and the paucity of comprehensive data on its epidemiology, clinical presentation, and treatment outcomes, a scoping review is warranted to collect the available literature on tongue melanoma, focusing on its prev-alence, risk factors, clinical features, and treatment strategies. By synthesizing these findings our study aims to identify existing gaps in the current body of knowledge to aid the planning of future research to improve the diagnosis and management of this rare and challenging condition.” and, in the Materials: “This study provides a scoping review of research on malignant melanoma and clear cell sarcoma of the tongue to determine what is the available data on the epidemiology, clinical features, histopathological characteristics, treatment strategies, and outcomes of this malignancy.”
- Use more recent PRISMA criteria, in which there are some modifications of those referred to (Stroup 2000, Moher 2009), such as the flow chart and the different checklists.
Thank you, we have also clarified this aspect in the Materials: “This study was conducted following the Preferred Reporting Items for Systematic Reviews and Meta-Analyses (PRISMA) extension for scoping reviews (PRISMA-ScR) [22].”
- Tricco, A.C.; Lillie, E.; Zarin, W.; O’Brien, K.K.; Colquhoun, H.; Levac, D.; Moher, D.; Peters, M.D.J.; Horsley, T.; Weeks, L.; et al. PRISMA Extension for Scoping Reviews (PRISMA-ScR): Checklist and Explanation. Ann Intern Med 2018, 169, 467–473, doi:10.7326/M18-0850.
Please also see, the revised flowchart in “Figure 2. Flow chart of search results and study selection according to the Preferred Reporting Items for Systematic Reviews and Meta-Analyses (PRISMA) extension for scoping reviews (PRISMA-ScR).” And the relevant checklist in the supplementary materials.
- Define the quality of the studies. If the reviewed articles offer quality related to the guideline for publishing cases (Clinical Case Reporting Guideline)
Thank you for your comment. We have added a relevant statement to the methods section, as well as to the Limitations
In the Methods: “This review adheres to the framework of a scoping review, focusing on mapping the existing evidence on tongue melanoma. Consistent with the objectives of scoping reviews, no formal critical appraisal of sources was done to provide a broader overview of the existing evidence [23,24].”
23.Munn, Z.; Peters, M.D.J.; Stern, C.; Tufanaru, C.; McArthur, A.; Aromataris, E. Systematic Review or Scoping Review? Guidance for Authors When Choosing between a Systematic or Scoping Review Approach. BMC Med Res Methodol 2018, 18, 143, doi:10.1186/s12874-018-0611-x.
24.Peters, M.D.J.; Godfrey, C.M.; Khalil, H.; McInerney, P.; Parker, D.; Soares, C.B. Guidance for Conducting Systematic Scoping Reviews. Int J Evid Based Healthc 2015, 13, 141–146, doi:10.1097/XEB.0000000000000050.
And, in the Limitations: “Consequently, we did not aim to define the quality of the included studies, which con-sisted in case reports and small case series.”
- Selection of articles with inclusion criteria of melanoma vs clear cell sarcoma diagnosis.
Thank you, we have added a relevant statement to the Methods section.
Please see: “Studies were selected based on predefined inclusion and exclusion criteria. We included case reports, case series, and review articles that provided information on clinical presentation, histopathology, treatment, or outcomes of tongue melanoma. To ensure inclusivity, case reports with incomplete data were also included, recognizing the rarity of this condition and the need to synthesize as much evidence as possible. Studies were excluded if they lacked relevance to tongue melanoma, or did not provide clinical or histopathological data.”
- Include some case of tongue melanoma described in previous case series.
Thank you, specific case details from the reviewed articles are detailed in “Table S2: List of publications included in the review.”
- It would be necessary to include the table of the reviewed articles in the manuscript as well as the corresponding bibliographic references so that readers can keep in mind their characteristics.
Thank you for your comment. This has been provided in the supplementary materials. Please see “Table S2: List of publications included in the review” were data charted from each study is reported.
In summary,
1) the cases must be presented in a transparent manner and diagnosed with melanoma, completing the entire sample [Spindle cell (8), epitheloid cell (3) mixed (8)....];
Thank you, this has been detailed in paragraph 3.4 Histopathology. Please see: “Though all cases were histologically confirmed (melanoma in 43 cases and CCS in 4), a description of histopathological findings was reported for 23 cases. …”
2) set objectives and conclusions accordingly.
Thank you for your constructive comment. The objectives of this scoping review are now clearly stated at the end of the introduction paragraph and the conclusions paragraph has been thoroughly rewritten to improve clarity. We have also added a relevant limitations paragraph.
Please see: “Given the rarity of melanoma of the tongue and the paucity of comprehensive data on its epidemiology, clinical presentation, and treatment outcomes, a scoping review is warranted to collect the available literature on tongue melanoma, focusing on its prev-alence, risk factors, clinical features, and treatment strategies. By synthesizing these findings our study aims to identify existing gaps in the current body of knowledge to aid the planning of future research to improve the diagnosis and management of this rare and challenging condition.”
And, in the Conclusions: “Tongue melanoma, though exceedingly rare, is a malignancy of considerable clinical importance due to its aggressive nature and challenging diagnosis. Its rarity demands heightened awareness among clinicians, as early-stage presentations often mimic benign or less aggressive lesions, leading to delayed diagnoses. A definitive diagnosis hinges on meticulous histopathological examination supported by immunohistochemical markers, which are essential for distinguishing tongue melanoma from other oral neoplasms. Despite advances in oncology, the prognosis for tongue melanoma remains poor, with high rates of recurrence and metastasis. Traditional therapies such as radiotherapy and chemotherapy show limited efficacy, while novel immunotherapies, particularly PD-1/PD-L1 inhibitors, hold promise but require further specific validation for mucosal melanomas. To address these challenges, a comprehensive and multidisciplinary strategy is essential. This includes not only medical and surgical interventions but also supportive care to manage the functional impairments commonly associated with treatment, such as difficulties in speech and swallowing. Advances in reconstructive techniques and post-operative care have the potential to significantly enhance patients' quality of life. Finally, basic research to unravel the underlying genetic and molecular mechanisms will be critical to improve our understanding on the pathophysiology of this rare entity, bridging the current knowledge gap in tongue melanoma.”
Thank you
Reviewer 3 Report
Comments and Suggestions for Authors
The authors proposed an interesting subject - a review of malignant melanoma of the tongue. They found 47 cases in the literature data, made a descriptive statistic, and analyzed the results. This review is based on 30 references.
Introduction
The introduction is short and refers only to malignant melanoma of the tongue and closely related entity (CCS).
The authors are invited to provide definitions of malignant melanoma and differentiate both forms (cutaneous and mucosal).
Then, they could classify the head and neck mucosal melanoma and its locations. They could refer to CCS and case numbers in the literature data, showing the differences between both malignancies.
They can describe oral melanomas (achromic and pigmented) and indicate their differences.
Then, they can indicate general data regarding evolution, treatment, and prognostic.
At the end of the introduction, they can show the reason for selecting this subject for review, the novelty of the presentation compared to previous ones, and how their work enriches the current scientific database.
Materials and methods
Please clearly show what "from inception" means.
Results:
For better understanding, please show a table with recorded data after descriptive analysis (frequency and relative frequency) and p-values in each section. Then, comment all data in the MS text.
For relevance and reader interest, please show some images with tumor localization, histological examination, etc
Then, all data regarding evolution, treatment, prognostic, recurrences, etc., will be correlated with sociodemographics and other comorbidities and bad habits if they exist.
Discussion
Please separate some subsections in the Discussion and provide better comments on the results.
Please show at the end
More references should be added.
After all, please revise the conclusions and also show future perspectives.
Please edit the references in MDPI style.
Author Response
Reviewer 3
The authors proposed an interesting subject - a review of malignant melanoma of the tongue. They found 47 cases in the literature data, made a descriptive statistic, and analyzed the results. This review is based on 30 references.
Dear Reviewer 3,
Thank you for your careful review and for your valuable suggestions. We have carefully considered each issue raised and our response are provided point by point below.
Introduction
The introduction is short and refers only to malignant melanoma of the tongue and closely related entity (CCS).
Thank you for your constructive comments, the introduction has been expanded in accordance with your suggestions. We have included additional information on the classification, clinical presentation, management, and prognosis of mucosal melanomas in the head and neck region, with a particular focus on oral melanoma and its various clinical forms (especially the amelanotic variant). Additional insights on clear cell sarcoma have also been provided. To enhance clarity, we have included an illustrative image of the molecular mechanisms involved. Furthermore, the aims of the study have been reformulated in the concluding section of the introduction.
The authors are invited to provide definitions of malignant melanoma and differentiate both forms (cutaneous and mucosal).
Please see in the text: “Primary mucosal malignant melanoma of the head and neck (HNMM) is a rare and aggressive neoplasm, with the tongue being an especially uncommon site for its occurrence. HNMM arises from melanocytes located in the basal layer of the mucosa across various anatomic sites [1]. Unlike cutaneous melanoma, which predominantly develops in the epidermis and is linked to ultraviolet (UV) radiation exposure, mucosal melanoma differs significantly in its epidemiology, genetic profile, clinical behavior, and response to therapy [1,2]. It is not induced by UV radiation and is associated with a poorer prognosis and lower response rates to immunotherapy compared to cutaneous melanoma. … Oral melanoma lacks most equivalent histological landmarks of cutaneous melanoma; instead, tumor thickness or volume serves as a prognostic indicator.”
Then, they could classify the head and neck mucosal melanoma and its locations. They could refer to CCS and case numbers in the literature data, showing the differences between both malignancies.
Please see in the text: “HNMM can involve all the aerodigestive tract, including the oral cavity, the pharyn-go-larynx, the nasal fossa, and the paranasal sinuses. Mucosal malignant melanomas of the head and neck most commonly originate in the sinonasal region, accounting for up to two-thirds of cases [3]. These tumors primarily arise from the lateral nasal wall and nasal septum. Melanomas of the oral cavity, which constitute approximately one-fourth of cases, frequently affect the palate or gingiva, are often diagnosed at advanced stages, and ac-companied by lymph node metastases. Histologically, these melanomas exhibit diverse growth patterns, including spindled, perivascular, and solid forms [2,3]. … CCS of the tongue differs from melanoma in its presentation, typically appearing as a rapidly growing, mobile nodular mass or a submucosal swelling [20].”
They can describe oral melanomas (achromic and pigmented) and indicate their differences.
Please see in the text: “Amelanotic mucosal melanoma displays diverse histological patterns, including epithe-lioid, spindle, pleomorphic, small cell, and plasmacytoid variants. The absence of melanin pigment complicates diagnosis [15]. Differential diagnoses include … Other prognostic factors include pigmentation levels and cell type, with nonepithelioid cell types and strong pigmentation associated to better survival outcomes.”
Then, they can indicate general data regarding evolution, treatment, and prognostic.
Please see in the text: “Oral melanomas are often asymptomatic until advanced stages, as patients rarely examine their oral cavity closely. They are usually diagnosed after significant swelling, tooth mobility, or bleeding prompts medical attention. Lesions range from 1.0 mm to over 1.0 cm, with reports of pre-existing pigmented lesions suggesting undiagnosed radial growth-phase melanomas [4]. Treatments typically involve surgery, interferon alpha-2b, immune checkpoint inhibitors, BRAF and MEK inhibitors, and radiotherapy. Multimodal therapy provides the best chance for relapse-free survival compared to single-treatment approaches [5]. The prognosis is poor, with a 5-year survival rate of 10-25%. Early detection and radical surgical excision may significantly improve outcomes. … Late-stage diagnosis often correlates to extensive tumors and metastases, with most pa-tients succumbing to the disease within 2 years. Recurrence and metastasis are common post-surgery. Survival rates vary widely in the literature, ranging from 4.5% to 48%, with most data clustering between 10-25% [6].”
At the end of the introduction, they can show the reason for selecting this subject for review, the novelty of the presentation compared to previous ones, and how their work enriches the current scientific database.
Thank you for your constructive comment. The objectives of this scoping review are now clearly stated at the end of the introduction paragraph to improve clarity.
Please see: “Given the rarity of melanoma of the tongue and the paucity of comprehensive data on its epidemiology, clinical presentation, and treatment outcomes, a scoping review is warranted to collect the available literature on tongue melanoma, focusing on its prev-alence, risk factors, clinical features, and treatment strategies. By synthesizing these findings our study aims to identify existing gaps in the current body of knowledge to aid the planning of future research to improve the diagnosis and management of this rare and challenging condition.”
Materials and methods
Please clearly show what "from inception" means.
Thank you, the sentence was rephrased to avoid a potential source of confusion: “A literature search was conducted from 1941 (year of the first documented case) [25] to 2024 (literature search performed on November 11, 2024),”
Results:
For better understanding, please show a table with recorded data after descriptive analysis (frequency and relative frequency) and p-values in each section. Then, comment all data in the MS text.
Thank you, descriptive analysis of data is summarized in the Supplementary materials: “Table S1: Summary of demographic, clinical, and histological characteristics of 47 cases of tongue melanoma”. It was not possible to perform additional analysis on these data and this limitation to our scoping review is clearly presented in paragraph 4.4 Limitations. Please see “… This scoping approach prevented us from performing additional statistics, including analysis of demographic data or survival.”
For relevance and reader interest, please show some images with tumor localization, histological examination, etc
Thank you for your constructive comment. We have added “Figure 1. Molecular pathways contributing to the progression of mucosal melanoma. Red-filled rectangles represent genes with activating mutations or amplifications, while blue-filled rectan-gles denote genes with inactivating mutations or deletions.” and “Figure 3. Frequency of different anatomical locations for melanoma of the tongue.” to the revised manuscript.
Then, all data regarding evolution, treatment, prognostic, recurrences, etc., will be correlated with sociodemographics and other comorbidities and bad habits if they exist.
Thank you for your suggestion. We considered that additional statistics were not applicable to our study as they are usually not done in scoping reviews. This limitation to our scoping review is clearly presented in paragraph 4.4 Limitations. Please see “… This scoping approach prevented us from performing additional statistics, including analysis of demographic data or survival.”
Discussion
Please separate some subsections in the Discussion and provide better comments on the results.
Thank you for your constructive comment. The discussion section has been enriched, particularly with additional insights into anti-PD1/PDL1 therapy, and has been organized into distinct subsections for improved readability.
Please show at the end
More references should be added.
Relevant references have been added to complement those already present, including citations of the cases included in the review (58 references in total).
After all, please revise the conclusions and also show future perspectives.
Thank you for your constructive comment. The conclusions paragraph has been thoroughly rewritten to improve clarity. We have also added a relevant limitations paragraph.
Please see: “Tongue melanoma, though exceedingly rare, is a malignancy of considerable clinical importance due to its aggressive nature and challenging diagnosis. Its rarity demands heightened awareness among clinicians, as early-stage presentations often mimic benign or less aggressive lesions, leading to delayed diagnoses. A definitive diagnosis hinges on meticulous histopathological examination supported by immunohistochemical markers, which are essential for distinguishing tongue melanoma from other oral neoplasms. Despite advances in oncology, the prognosis for tongue melanoma remains poor, with high rates of recurrence and metastasis. Traditional therapies such as radiotherapy and chemotherapy show limited efficacy, while novel immunotherapies, particularly PD-1/PD-L1 inhibitors, hold promise but require further specific validation for mucosal melanomas. To address these challenges, a comprehensive and multidisciplinary strategy is essential. This includes not only medical and surgical interventions but also supportive care to manage the functional impairments commonly associated with treatment, such as difficulties in speech and swallowing. Advances in reconstructive techniques and post-operative care have the potential to significantly enhance patients' quality of life. Finally, basic research to unravel the underlying genetic and molecular mechanisms will be critical to improve our understanding on the pathophysiology of this rare entity, bridging the current knowledge gap in tongue melanoma.”
Please edit the references in MDPI style.
Thank you, all citations have been formatted according to the MDPI style.
Reviewer 4 Report
Comments and Suggestions for Authors
Please, see the attached document.

Author Response
Reviewer 4
This manuscript by Di Guarda et al. analyzes the clinical and histopathological
characteristics of malignant melanoma of the tongue, a rare and aggressive
neoplasm. The study synthesizes data from 47 cases, emphasizing the challenges of
early diagnosis, high rates of metastasis, and poor prognosis. It emphasizes the need
for multidisciplinary management and further investigation of genetic mechanisms to
improve outcomes for this malignancy. The manuscript addresses an important topic
and presents valuable data, but significant improvements in statistical rigor, clarity,
and discussion of novel treatments are needed to maximize its impact.
I have included below a few suggestions and questions that could improve the quality
of the manuscript:
Dear Reviewer 4,
Thank you for your efforts on our manuscript and for your constructive suggestions. Each issue raised has been carefully considered and our responses are provided below point by point:
- Can you clarify the criteria for inclusion or exclusion of cases, especially regarding
case reports with incomplete data?
Thank you, we have added a relevant statement to the Methods section.
Please see: “Studies were selected based on predefined inclusion and exclusion criteria. We included case reports, case series, and review articles that provided information on clinical presentation, histopathology, treatment, or outcomes of tongue melanoma. To ensure inclusivity, case reports with incomplete data were also included, recognizing the rarity of this condition and the need to synthesize as much evidence as possible. Studies were excluded if they lacked relevance to tongue melanoma, or did not provide clinical or histopathological data.”
- Why was no formal assessment of bias performed for the included studies? Why
was an assessment of bias omitted, and how might this have affected the findings?
Thank you for your comment. We have added a relevant statement to the methods section, as well as to the Limitations
In the Methods: “This review adheres to the framework of a scoping review, focusing on mapping the existing evidence on tongue melanoma. Consistent with the objectives of scoping reviews, no formal critical appraisal of sources was done to provide a broader overview of the existing evidence [23,24].”
23.Munn, Z.; Peters, M.D.J.; Stern, C.; Tufanaru, C.; McArthur, A.; Aromataris, E. Systematic Review or Scoping Review? Guidance for Authors When Choosing between a Systematic or Scoping Review Approach. BMC Med Res Methodol 2018, 18, 143, doi:10.1186/s12874-018-0611-x.
24.Peters, M.D.J.; Godfrey, C.M.; Khalil, H.; McInerney, P.; Parker, D.; Soares, C.B. Guidance for Conducting Systematic Scoping Reviews. Int J Evid Based Healthc 2015, 13, 141–146, doi:10.1097/XEB.0000000000000050.
And, in the Limitations: “Consequently, we did not aim to define the quality of the included studies, which con-sisted in case reports and small case series.”
- Were any genetic studies (e.g., molecular profiling) of the cases considered?
A single study included genetic profiling of a tongue melanoma within a case series of oral melanomas. However, the results of this study provide a general interpretation of the genetic landscape of oral melanomas as a whole, rather than offering specific insights into tongue melanoma compared to other types of oral melanomas (Tanaka N, Odajima T, Mimura M, Ogi K, Dehari H, Kimijima Y, Kohama G. Expression of Rb, pRb2/p130, p53, and p16 proteins in malignant melanoma of oral mucosa. Oral Oncol. 2001 Apr;37(3):308-14. doi: 10.1016/s1368-8375(00)00107-x.). Please see in the text, in paragraph 4.1 Pathophysiology of tongue melanoma: “Genetic predisposition may play a significant role in the pathogenesis of tongue melanoma. Recent studies on oral malignant melanomas have identified mutations in tumor suppressor genes such as BAP1, as well as alterations in pathways involving p16 and p53 [49]. Loss of pRb2/p130 expression has been associated with more aggressive disease and poorer outcomes, whereas its presence may correlate with earlier malignancy stages [49].”
- Can you elaborate on why specific treatment modalities (e.g., immunotherapy)
were underrepresented in the results? Please include a discussion of new treatments
(e.g., immunotherapy) and their relevance to tongue melanoma.
Thank you, the underrepresentation of treatments such as immune checkpoint inhibitors (anti-PD1/PD-L1 and anti-CTLA4 therapies) in the data likely stems from the fact that the reviewed studies frequently predated the widespread clinical adoption of these agents. Additionally, several articles lacked comprehensive treatment descriptions. Considerations on immunotherapy and targeted therapies have been added to the "Discussion" section. Please see section 4.3 Treatment: “… Immune checkpoint inhibitors, particularly anti-PD1/PD-L1 therapies, have trans-formed the management of advanced melanomas, including mucosal subtypes, by demonstrating superior survival outcomes compared to traditional therapies [47,53]. However, their role in tongue melanoma specifically remains unclear due to the rarity of this condition and the limited representation of these agents in the reviewed cases. This underrepresentation is partly attributed to the retrospective nature of most studies and their publication before the widespread clinical adoption of these agents. Despite these limitations, the distinct molecular and clinical features of tongue melanoma—such as its high rates of locoregional and distant metastasis—suggest that anti-PD1 therapies could play a pivotal role in improving outcomes. Nevertheless, robust conclusions are ham-pered by the lack of prospective, tongue-specific studies and incomplete reporting of subsequent lines of therapy in many cases. Targeted therapies, including BRAF inhib-itors (e.g., vemurafenib, dabrafenib) and MEK inhibitors (e.g., trametinib), have shown significant efficacy in BRAF-mutated cutaneous melanomas [54]. However, BRAF mu-tations are rare in mucosal melanomas, including tongue melanoma, limiting the ap-plicability of these treatments. In contrast, c-KIT mutations are more frequently observed in mucosal melanomas, particularly in the head and neck region [55,56]. KIT inhibitors, such as imatinib, have demonstrated modest efficacy in KIT-mutated cases, with isolated reports of durable responses. The combination of KIT inhibitors with immune checkpoint inhibitors is a potential therapeutic avenue that warrants further exploration, given the promising synergistic effects observed in other melanoma subtypes. Furthermore, emerging strategies, such as neoadjuvant and combination immunotherapies (e.g., an-ti-PD1 with anti-CTLA4), hold promise in managing locally advanced or metastatic mucosal melanomas [57]. While combination therapies have demonstrated improved progression-free survival and response rates in cutaneous melanoma, their benefits in mucosal melanomas, including tongue melanoma, appear less pronounced and are as-sociated with higher toxicity. The lack of statistically significant differences in compar-ative analyses between monotherapy and combination regimens for mucosal melanomas, as well as the increased risk of severe adverse effects, suggests that anti-PD1 monotherapy may currently be the most prudent first-line treatment for tongue melanoma [53]. Given the paucity of data specific to tongue melanoma, there remains an urgent need for further research to elucidate the impact of these therapies. Prospective studies investigating the molecular profile of tongue melanoma, particularly the predictive value of PD-L1 ex-pression and KIT mutations, are essential to guide the selection of targeted drugs and immunotherapies.”
- How did you ensure the reliability and consistency of data extraction across
studies? Were alternative methods (e.g., meta-analysis) considered for survival and
treatment efficacy analysis?
Thank you for your suggestion. Please see, in the Methods section: “The process of selection of the sources of evidence was performed by two authors (ADG and AS) that independently assessed the titles, abstracts, and text of studies identified by the search; results were compared and discrepancies were resolved by discussions, in-cluding a third author (CC) when needed. Then, charting of data from the included sources was performed independently by two reviewers (ADG and AS) using a stand-ardized data-charting template to capture variables such as patient demographics, clinical presentation, histopathology, treatment modalities, and outcomes. While efforts were made to include comprehensive data, many studies lacked long-term follow-up infor-mation, precluding insights into overall survival and the use of second-line or alternative therapeutic approaches. In several cases reported by pathologists, only clinical descrip-tions and diagnoses were provided, with no data on follow-up or treatment outcomes.”
Moreover, we considered that additional statistics were not applicable to our study as they are usually not done in scoping reviews. This limitation to our scoping review is clearly presented in paragraph 4.4 Limitations. Please see “… This scoping approach prevented us from performing additional statistics, including analysis of demographic data or survival.”
- Can you clarify whether newer therapeutic modalities such as checkpoint inhibitors
have been systematically reviewed? While immune checkpoint inhibitors (e.g., PD-
1/PD-L1 therapies) and targeted therapies (e.g., BRAF/MEK inhibitors) are briefly
mentioned, their relevance is not thoroughly reviewed in the context of mucosal
melanomas.
Thank you for your comment, novel therapeutic modalities together with other treatment strategies are included in the data items that were charted in this scoping review (please see the Methods section: “This study provides a scoping review of research on malignant melanoma and clear cell sarcoma of the tongue to determine what is the available data on the … treatment strategies, and outcomes of this malignancy.”). Moreover, relevant considerations on immunotherapy and targeted therapies have been added to the "Discussion" section (please see response to comment 4) and to the Conclusions (“Traditional therapies such as radiotherapy and chemotherapy show limited efficacy, while novel immunotherapies, particularly PD-1/PD-L1 inhibitors, hold promise but require further specific validation for mucosal melanomas.”).
- The lack of statistical comparison between different treatment modalities
undermines the strength of conclusions about their efficacy. The review includes 47
cases, but given the rarity of tongue melanoma, more robust conclusions would
require a larger data set or a pooled analysis.
Thank you for your comment, a Limitations paragraph has been added to highlight the limitations of our scoping approach as well as those of the included studies. Please see: “The scoping methodology of this review carries inherent limitations: charted data was reported in the review irrespective of a critical appraisal of the sources in order to achieve a comprehensive summary of all existing evidence. Consequently, we did not aim to define the quality of the included studies, which consisted in case reports and small case series. Moreover, the paucity of evidence found suggested against undertaking a sys-tematic review. This scoping approach prevented us from performing additional statistics, including analysis of demographic data or survival. As far as the limitations of the in-cluded studies are concerned, reports of isolated cases are prone to selection bias, as they often emphasize rare, severe, or atypical cases, potentially skewing the representation of tongue melanoma's clinical spectrum. Reporting bias may further affect the findings, as published cases are likely to highlight positive or unusual outcomes, while publication bias may result in underrepresentation of inconclusive or negative data. Despite these limitations, the review provides a comprehensive synthesis of the existing evidence, offering valuable insights into this rare malignancy. To mitigate these biases in future research, setting up large disease registries or multicenter studies will be essential. Another significant limitation of the included studies was the frequent lack of long-term follow-up data, which restricted our ability to assess overall survival and therapeutic response beyond first-line treatments. This was further complicated by some reports which completely lacked follow-up and treatment details.”
- It is appropriate to add several figures that would clearly explain the mechanism of
development of this type of tumors, the histopathological picture, and the
mechanisms of current and prospective treatment.
Thank you for your constructive comment. We have added “Figure 1. Molecular pathways contributing to the progression of mucosal melanoma. Red-filled rectangles represent genes with activating mutations or amplifications, while blue-filled rectan-gles denote genes with inactivating mutations or deletions.” and “Figure 3. Frequency of different anatomical locations for melanoma of the tongue.” to the revised manuscript.
Round 2
Reviewer 1 Report
Comments and Suggestions for Authors
No further comments. All my previous suggestions have sufficiently been explained.
Author Response
Dear Reviewer,
Thank you for your constructive contribution to our manuscript.
Reviewer 2 Report
Comments and Suggestions for Authors
I only ask for two clarifications.
- In the scoping review, the number of studies included is 47, but you only mention [26–40], line 188. It should be explained.
- Possibly table “3” on line 230 should be table 2.
Thank you very much for the answers.
Author Response
Dear Reviewer,
Thank you for your further efforts on our manuscript and for your comments. We have modified the text accordingly:
- All 47 reviewed studies have now been included in the reference list and cited in the appropriate position in the text (please see references 13,14,17-20,25-64)
- We apologize for this mistake which has been corrected to “Table 2”.
Reviewer 3 Report
Comments and Suggestions for Authors
The reviewer highly appreciates the authors' efforts to revise the manuscript according to the previous review report. The reviewer noted that the revised version is substantially improved and has no additional comments.
Author Response
Dear Reviewer,
Thank you for your encouraging comments.
Reviewer 4 Report
Comments and Suggestions for Authors
Since significant corrections have been made, I recommend it for publication.
Author Response
Dear Reviewer,
Thank you for your careful revision of our paper and for your recommendation.